# Acidosis Activates the Nrf2 Pathway in Renal Proximal Tubule-Derived Cells through a Crosstalk with Renal Fibroblasts

**DOI:** 10.3390/antiox12020412

**Published:** 2023-02-08

**Authors:** Marie-Christin Schulz, Virginie Dubourg, Alexander Nolze, Michael Kopf, Gerald Schwerdt, Michael Gekle

**Affiliations:** Julius Bernstein Institute of Physiology, Magdeburger Straße 6, 06112 Halle (Saale), Germany

**Keywords:** cellular crosstalk, chronic kidney diseases, extracellular acidosis, Nrf2, transketolase, renal cell crosstalk

## Abstract

Crosstalk of renal epithelial cells with interstitial fibroblasts plays an important role in kidney pathophysiology. A previous study showed that crosstalk between renal epithelial cells and renal fibroblasts protects against acidosis-induced damage. In order to gain further mechanistic insight into this crosstalk, we investigated the effect of acidosis on the transcriptome of renal epithelial cells (NRK-52E) and renal fibroblasts (NRK-49F) in co-culture by RNASeq, bioinformatics analysis and experimental validation. Cells were exposed to acidic media or control media for 48 h. RNA and protein from whole cell lysate were isolated. In addition, cells were fractionated into cytosol, nucleus and chromatin. RNASeq data were analyzed for differential expression and pathway enrichment (ingenuity pathway analysis, IPA, QIAGEN). Total and phosphorylated protein expression was assessed by Western blot (WB). Transcription factor activity was assessed by luciferase reporter assay. Bioinformatic analysis using differentially expressed genes according to RNASeq (7834 for NRK-52E and 3197 for NRK-49F) predicted the antioxidant and cell-protective Nrf2 pathway as acidosis-induced in NRK-52E and NRK-49F cells. Activation of Nrf2 comprises enhanced Nrf2 phosphorylation, nuclear translocation, DNA binding and initiation of a cell protective transcriptional program. Our data show that acidosis enhances chromatin-associated Nrf2 expression and the abundance of phosphorylated Nrf2 in the chromatin fraction of NRK-52E cells in co-culture but not in monoculture. Furthermore, acidosis enhances the activity of a reporter for Nrf2 (ARE-luciferase). Despite the bioinformatics prediction, NRK-49F cells did not respond with Nrf2 activation. Transketolase (TKT) is an important regulator of antioxidant and homeostatic responses in the kidney and a canonical Nrf2 target gene. We show that protein and mRNA expression of TKT is increased in NRK-52E cells under co-culture but not under monoculture conditions. In conclusion, our data show that extracellular acidosis activates the cytoprotective transcription factor Nrf2 in renal epithelial cells co-cultivated with renal fibroblasts, thereby enhancing the expression of cytoprotective TKT. This protective response is not observed in monoculture. Activation of the Nrf2 pathway represents a co-operative cellular strategy of protection against acidosis.

## 1. Introduction

The global prevalence of chronic kidney disease (CKD) is around 12%, leading to 1.2 million deaths annually [1]. Therefore, it is crucial to understand the mechanisms which initiate the development and progression of CKD. There is evidence that the proximal tubule is often involved in the mediation of pathological events during the early stages of CKD [2]. Moreover, many hints show that the proximal tubule plays a key role for the progression of CKD [3]. Stressed proximal tubule cells can undergo an EMT to initiate tissue repair [4]. During this process, the proximal tubule cells can generate a proinflammatory milieu, supporting the formation of myofibroblasts [3].

Inflammation and ischemia as well as fibrosis are accompanied by extracellular acidosis [5,6], whereby proximal tubule cells and renal fibroblasts are exposed to pH values well below 6.5 in rodents [7,8,9]. Extracellular acidosis may act as an additional stimulus, promoting further pathophysiological changes, finally resulting in a vicious cycle.

Under physiological conditions, tubule epithelial cells and tissue fibroblasts can communicate via soluble mediators (e.g., cytokines and COX-2 metabolites) and matrix proteins [10,11,12,13,14]. This cellular crosstalk has an impact on the differentiation of epithelial cells, matrix protein formation and the composition of the interstitial micro milieu [15]. Pathological micro milieu changes, such as acidosis, can be cause and/or consequence of an altered cellular crosstalk, and thereby enhance the progression of pathological processes.

In a previous study, it was shown that extracellular acidosis causes an inflammatory and fibrotic response in cells derived from proximal tubule and renal fibroblasts. This was true as long as the cells were cultivated in monoculture. When the model was extended to a co-culture model, the acidosis-induced damage was prevented [16]. On the basis of these results, we developed the hypothesis that an epithelial–fibroblast communication protects the cells against acidosis-induced damage. The aim of the present study was to identify mechanisms that can explain the described protective effect of cellular communication.

The Nrf2 signaling pathway acts cytoprotectively in a variety of stress situations via an adaptation of the transcriptional program that, e.g., prevents damage from enhanced reactive oxygen formation. In the kidney, activation of the Nrf2 pathway prevents tubular damage progression after ischemia/reperfusion due to the upregulation of genes that regulate xenobiotic disposition, redox balance and the supply of NADPH and other cellular fuels [17,18,19,20]. Thus, in the kidney, Nrf2 regulates genes that co-ordinate homeostatic processes to prevent tissue damage.

In the inactive state, the transcription factor Nrf2 is located in the cytosol, bound by Keap1, and is continuously degraded by the proteasome. Activation of Nrf2 requires its dissociation from Keap1, translocation to the nucleus and phosphorylation. Different protein kinases, including PKC, CK2, AMPK, Plk2 and MAPK, mediate the phosphorylation of Nrf2 at different sites [20,21,22], whereat phosphorylation at Ser 40 is of special importance. In the nucleus, Nrf2 binds to chromatin via antioxidative response elements (AREs) in promotor regions and induces the transcription of protective enzymes such as peroxidases, reductases and transferases [23]. There are a few studies indicating an activation of the Nrf2 pathway by acidosis [24,25,26].

## 2. Materials and Methods

If not stated otherwise, chemicals were purchased from Sigma-Aldrich, Munich, Germany.

### 2.1. Cell Culture

Normal rat kidney fibroblasts (NRK-49F, ATCC CRL-1570) and normal rat kidney epithelial cells (NRK-52E, ATCC^®^ CRL-1571) were grown in DMEM supplemented with 5% fetal calf serum (FCS) and 2 g/L NaHCO_3_ at 37 °C under a humidified 5% CO_2_ atmosphere and subcultivated once per week before confluence.

### 2.2. Experimental Setup

NRK-52E cells were grown on permeable filter inserts (pore size 0.4 µm) and NRK-49F cells were grown in 6-well plates. When cells were confluent, the filter inserts, containing NRK-52E cells, were transferred to the 6-well plates, containing NRK-49F cells. This indirect co-culture was used for the following experiments. The co-culture was transferred to medium without additional FCS supplementation for 24 h and afterwards incubated with experimental conditions for 48 h. The aim of this study is to reveal mechanisms which are involved in the development of chronic kidney diseases. Therefore, we are committed to creating an experimental setting which approximates the pathological condition, as close as possible. This includes the longest possible incubation time, without critical decreases in cell viability. In a previous study [16], we showed that incubation for 48 h is suitable. In addition, to ensure comparability of the results, we used the same experimental conditions in the present study. Control cells were exposed to media with a pH value of 7.4 and the acidosis group was incubated with media at a pH value of 6.0. Extracellular pH (pHe) was measured with a blood gas analyzer (ABL5, Radiometer, Copenhagen, Denmark). Only a minor reduction in pHe of medium was observed during the chosen incubation periods. Therefore, the experiments could be performed under well-controlled conditions. Epithelial cells attach to the substrate via their basolateral membrane, whereas the apical membrane faces towards the lumen. In vivo, this substrate is the basement membrane, which separates the epithelial cell from the interstitial space that harbors fibroblasts. Thus, in order to mimic the in vivo situation properly, NRK-52E cells were cultivated on the membrane of a Transwell insert. NRK-49F cells were cultivated on the bottom of the wells of a 6-well-plate, in which the Transwells were placed. There was no direct contact of the membrane with the bottom of the well. Due to the pore size of the membrane (0.4 µm), a contamination of the upper (Transwell) compartment with fibroblasts or of the lower (6-well) compartment with epithelial cells is impossible. For processing at the end of the incubation period, the Transwells and the 6-well plate were separated, the lower side of the membrane as rinsed with ice-cold PBS and inspected to ensure that no material from the lower compartment was attached. Thereafter, the cells were processed separately. With these procedures, we avoided relevant crosscontamination of the two cell types. There is a remaining, very small, chance of a few fibroblasts adhering to the bottom of the Transwell membrane. However, this very low level of potential contamination is unlikely to change the results.

### 2.3. RNA Sample Preparation

Total RNA of the cells in co-culture was isolated with BlueZol Reagent as described in the user manual. Possible genomic DNA contaminations were removed with a “Turbo DNAse-free kit”, following the “rigorous DNAse treatment” protocol from the manufacturer. All samples were purified by ethanol precipitation (with 180 µL water, 18 µL 3 M sodium acetate, 4 µL of 20 mg/mL glycogen and 600 µL of 99.9% ethanol, final volume 802 µL + sample volume). The quality of the RNA samples was assessed using a 2100 Bioanalyzer (Agilent Technologies, Karlsruhe, Germany). All samples had an RNA integrity number (RIN) above 7 (with 10 as the maximum possible value).

### 2.4. RNA Sequencing

The RNA sequencing was performed by Novogene Co., Ltd. (Cambridge, UK). Four samples were used for each experimental group. The company carried out the library preparation (poly(A) enrichment), and the paired-end sequencing (2 × 150 bp) run on a NovaSeq6000 Illumina system. The service company provided adaptor clipping and data quality control as well. Read mapping to the rat genome rn6 was performed with tophat2 (v.2.0.14) [27], and featureCounts 2.0 (–p –M –t exon –g gene_id) [28] was used to count the mapped reads. Gene annotation was performed using BiomaRt (v.2.44.4) [20] to access the Ensembl (from jan2020.archive.ensembl.org, accessed on 20 January 2020).

### 2.5. Differential Expression Analysis and Functional Analysis

The differential expression analysis was performed using SARTools (1.6.0) [29], a DESeq2- (1.18.1) and edgeR-based (3.20.8) R pipeline [30,31]. Normalization factors were calculated with the “trimmed mean of M value” (TMM) method in the edgeR analysis. Significantly “differentially expressed genes” (DEGs) were defined as genes with a false discovery rate (FDR) below 0.01 in both DESeq2 and edgeR outputs (overlap of the respective results).

Ingenuity pathway analysis (IPA) software (Qiagen) was used for canonical pathway analysis and upstream regulator analysis on lists of significantly regulated genes [32]. The Ensembl identifiers of the latter were mapped to networks incorporated into the software database. The featured “Comparison Analysis” tool was used to match the different results. Results were filtered for |Z-score| ≥ 2 and adjusted (Benjamini–Hochberg) *p*-value ≤ 0.05.

### 2.6. Quantitative PCR

Total RNA from cells in co-culture was isolated with BlueZol Reagent (Serva, Germany) as described in the user manual. Reverse transcription of RNA was performed using a commercial kit from New England Biolabs (DNase I, M0303L, New England Biolabs GmbH, Frankfurt am Main, Germany) according to their instructions. Real-time PCR was performed using the SYBR Green reagent (SsoAdvanced Universal SYBR Green Supermix, 9207996, Bio-Rad, Munich, Germany). Primers for PCR were synthesized by Microsynth AG, Balgach, Switzerland. Primer sequences and annealing temperatures are given in Appendix A. Fold change of gene expression was calculated by the 2^ΔΔCt^ method. *HRPT* was used as reference.

### 2.7. Cell Fractionation

Cells were fractionated into a cytosolic, nuclear and chromatin fraction. First, the cells were suspended in PBS (Appendix A) + protease inhibitor cocktail (PIC, 1:500). The cells were centrifuged for 5 min, at 400× *g*. The supernatant was removed and the cell pellet was resuspended in 180 µL equilibration buffer (Appendix A) + PIC (1:100). The solution was centrifuged for 1 min, at 14,000× *g* and the supernatant was removed. Fifty microliters of lysis buffer (Appendix A) + PIC (1:100) was added to the pellet and incubated for 10 min. The mixture was centrifuged for 5 min, at 14,000× *g*.

The supernatant contained cytosolic proteins and was concentrated with the help of Amicon^®^ Ultra-4 10K centrifugal filter devices (Sigma-Aldrich, Munich, Germany). The pellet contained nuclear and chromatin protein and was incubated with 60 µL extraction buffer + PIC (1:100) for 30 min. The mixture was centrifuged for 5 min, at 14,000× *g*. The supernatant contained nuclear proteins and the chromatin proteins remained in the pellet. The pellet was purified with PBS + PIC (1:100) and centrifuged for 5 min, at 14,000× *g*. Finally, the pellet was resuspended in 50 µL MOPS triton (Appendix A) + benzonase (250 U/mL). It was incubated for 30 min and ultrasonically lysed two times. The samples were mixed with 16.6% of total volume 6 × Laemmli buffer (Appendix A) and used for Western blotting described in Section 2.8.

### 2.8. Western Blot

Cells were lysed in 50 µL ice-cold MOPS Triton buffer (Appendix A). Afterwards, cells were centrifuged at 14,000× *g* for 10 min (4 °C) and added to 16.6% of total volume 6 × Laemmli buffer (Appendix A). Before loading the gel, samples were heated to 95 °C for 5–10 min. Proteins were separated by 12% sodium dodecyl sulfate–polyacrylamide gel electrophoresis (SDS-PAGE) and transferred onto a nitrocellulose membrane. The proteins on the nitrocellulose membrane were stained with Ponceau S solution (AppliChem GmbH, Darmstadt, Germany). We took a picture of the stained membrane with the Bio-Rad ChemiDoc™ XRS gel documentation system (Bio-Rad Laboratories GmbH, Feldkirchen, Germany). After staining, the membrane was blocked with 5% nonfat dry milk powder in TRIS-buffered saline with Tween20 (TBS Tween20) (Appendix A) and incubated with primary antibody (Table 1) diluted in 5% bovine serum albumin (BSA) in TBS Tween20 overnight. After removing the primary antibody and washing the membrane, a secondary antibody coupled to horseradish peroxidase was diluted 1:1000 in 5% nonfat dry milk powder in TBS Tween20 and added to the membrane. After removal of the secondary antibody solution, three wash steps in TBS Tween20 were performed. Finally, the membrane was incubated for 5 min with Clarity™ Western ECL Substrate (Bio-Rad, Munich, GER) and the peroxidase activity-based light emission was recorded by an imaging system (Image Quant LAS4000, GE Health care, Buckinghamshire, GB). Alternatively, IRDye-coupled fluorescent secondary antibodies (1:20,000 in 5% nonfat dry milk powder in TBS Tween20; LI-COR, Biosciences, Lincoln, NE, USA) were used and visualized with the Odyssey infrared imaging system (LI-COR, Biosciences). Density of the total protein, made visible with Ponceau S solution and protein target bands, was quantified using Quantity One software, version 4.6.9 (Bio-Rad, Munich, Germany). The normalization was performed with the Ponceau S-stained total protein. After detecting pNrf, the membranes were incubated for 15 min at 65 °C with restore™western blot stripping buffer (Thermo Fisher Scientific GmbH, Dreieich, Germany) to remove the antibodies. Afterwards the membranes were reused to detect total Nrf2.

### 2.9. ARE-Luciferase Assay

Cells were transfected with a pGL4.37-luc2p/ARE/Hygro vector (Promega Corporation, Madison, WI, USA). After 24 h, the media were removed and cells were put together as a co-culture and incubated for 48 h with acidic media. After the incubation, cells were lysed, and luciferase activity in the lysate measured using the Dual Luciferase Assay System (Promega, Madison, WI, USA). Beta-galactosidase activity was measured at 405 nm by colorimetric assay with ortho-nitrophenyl-galactosidase as substrate [33]. The luciferase activity was normalized to the beta-galactosidase activity.

### 2.10. Data Analysis

All data are given as mean ± SEM. Statistical significance was determined by unpaired Student’s *t*-test or rank sum, as appropriate. Differences were considered statistically significant when *p* < 0.05.

## 3. Results

### 3.1. Impact of Extracellular Acidosis on General Gene Expression

RNA sequencing followed by differential expression and pathway analysis was performed to identify possible candidates involved in the acidosis effect. Figure 1a and Appendix A show the result of clustering according to fragments per million mapped fragments (FPM) values. The sample groups were clearly separated according to cell type and pH (see also similarity matrix in Appendix A). Next, we identified differently expressed genes by DESeq2 and edgeR. Only genes identified by both algorithms were considered for further analysis. In NRK-52E cells, acidosis led to a change in the expression of 7834 genes (Figure 1b) and in NRK-49F cells to a change in the expression of 5314 genes (Figure 1c); 3197 genes were affected in both cell types (Figure 1d). Expression of the majority of genes affected in both cell types was changed in the same direction (Figure 1e). However, there is a substantial portion of genes which showed opposite regulation. Thus, there is a certain cell type specificity regarding the response to acidosis.

### 3.2. Genes Regulated by Extracellular Acidosis in Both Cell Lines Are Enriched in the Nrf2 Pathway

We applied the IPA^®^ software for in-depth pathway analysis. IPA annotated 3606 genes for NRK-52E and 1624 genes for NRK-49F (Figure 2a). Moreover, Figure 2a shows that 40% (NRK-52E) and 45% (NRK-49F) of affected genes were upregulated. *Canonical pathway analysis* with IPA predicts potential pathways responsible for the observed changes in gene expression. *Upstream analysis* by IPA provides predictions regarding mechanisms or signaling events upstream of the transcriptomic changes. Both types of analysis provide a measure of the predicted strength and direction of the effect (z-score) as well as a measure for the probability of false positive associations (Benjamini–Hochberg (B-H) adjusted *p*-value). We started with a canonical pathway analysis, using the list of the filtered genes (Figure 2a), an adjusted B-H *p*-value of 0.05 and a |z-score| ≥ 2. For both cell lines, the Nrf2-mediated oxidative stress response pathway was predicted to be activated with the highest scores (Figure 2b,c). For NRK-52E cells, the z-score was 2.5, and for NRK-49F cells it was 2.2. Subsequently, we applied the upstream analysis tool on the same set of genes and increased the stringency by setting the adjusted B-H *p*-value to 0.001. For NRK-52E cells, Nrf2 (NFE2L2) was predicted as an upstream regulator with a z-score of 2.68 and an adjusted B-H *p*-value < 10^−^^6^ (Figure 3). By contrast, Nrf2 was not identified as an upstream regulator for NRK-49F cells (Figure 3).

### 3.3. Impact of Extracellular Acidosis on Nrf2 Expression

Activation of the Nrf2 pathway can result from enhanced expression, phosphorylation or translocation to the nucleus. According to the RNA sequencing results, Nrf2 expression is enhanced by acidosis in NRK-52E but not in NRK-49F cells (Figure 4a,e) under co-culture conditions. Thus, we analyzed Nrf2 expression at the protein level after cell fractionation. Figure 4b,f show that a substantial fraction of Nrf2 protein is chromatin-associated, with a low amount of intact Nrf2 protein in the cytosol. Acidosis enhanced the expression of Nrf2 in NRK-52E cells (enhanced abundance in the chromatin fraction with no measurable changes in the other fractions) but not in NRK-49F cells (Figure 4b,c,f,g). In NRK-49F cells, we observed even a trend to reduced Nrf2 expression under acidotic conditions in co-culture. These results confirm the data on mRNA expression. Figure 4i–j show furthermore that acidosis exerted no effect on Nrf2 expression of NRK-52E cells in monoculture. Thus, cellular crosstalk is required for this effect, by a yet unknown mechanism.

### 3.4. Impact of Extracellular Acidosis on Nrf2 Phosphorylation and Activation

As mentioned above, activation of Nrf2 may comprise altered phosphorylation. To investigate the effect of acidosis on phosphorylation, expressions of phosphorylated Nrf2 (pNrf2) protein in cytosolic, nuclear and chromatin fractions were compared.

As shown in Figure 5a–c,e–g, acidosis enhanced the amount of pNrf2 associated with chromatin significantly in NRK-52E cells but not in NRK-49F cells. In NRK-49F cells, the pNrf2 signal in the chromatin fraction was in general very weak and difficult to detect, indicating a lower phosphorylation level, and decreased slightly under acidotic conditions. Due to the low expression level of pNrf2 in NRK-49F, the effect of acidosis must be handled with care. Yet, as described below, reporter gene analysis, as a second and independent technique to determine acidosis effects on Nrf2 activity (Figure 5h), confirmed that Nrf2 is not stimulated in NRK-49F cells.

Normalization of the acidosis-induced changes in Nrf2 phosphorylation in NRK-52E cells with the changes in total Nrf2 expression results in a ratio of 1.83 (median) with a 95% confidence interval [1.35; 2.33], indicating enhanced Nrf2 phosphorylation in addition to enhanced expression (Figure 5c,g).

Figure 5d shows the increase in transcriptional Nrf2 activity in NRK-52E cells during acidosis. Thus, in NRK-52E cells under co-culture conditions, acidosis-induced activation of the Nrf2 pathway finally leads to enhanced transcriptional activity. The Nrf2 reporter signal in NRK-49F cells was not enhanced by acidosis, in accordance with the results for Nrf2 expression and phosphorylation.

### 3.5. Acidosis Leads to Enhanced Expression of the Canonical Nrf2 Target Gene TKT

As a proof of principle, we determined the effect of acidosis on the expression of a canonical Nrf2 target gene with pathophysiological relevance for the kidney, transketolase (TKT). TKT is an important enzyme regulating the activity of oxidative and nonoxidative pentose phosphate pathways with the objective to protect cells from reactive oxygen species and to promote proliferation. TKT has also been proposed as a relevant factor for nephroprotection [18,19]. Figure 6a–c show the increase in TKT mRNA and protein expression in NRK-52E in co-culture under acidotic conditions.

When we performed the experiment under monoculture conditions, acidosis no longer stimulated TKT protein expression (Figure 6d,e), although there was still a slight increase in TKT mRNA (Figure 6f) that did not reach the level under co-culture (Figure 6c).

In summary, our results show that metabolic acidosis leads to an activation of the Nrf2 pathway in renal epithelial cells but not in fibroblasts in co-culture. The crosstalk between the two cell types is necessary for this cytoprotective cell-specific effect in NRK-52E cells. Furthermore, activation of the Nrf2 pathway translates into enhanced expression of Nrf2 target genes, shown as a proof of principle, for TKT. At present, we know that the crosstalk is necessary, but we do not know by which means the crosstalk occurs. This important question will be addressed in future studies.

## 4. Discussion

Following acute kidney injury and during chronic kidney diseases, proximal tubule cells interact with interstitial fibroblasts in a way that can lead to self-sustaining vicious cycles, promoting inflammation, fibrosis, ischemia and acidosis [3]. Acidosis is one of the nonspecific milieu factors that play a role in various pathophysiological settings supporting chronic vicious cycles that lead to inflammation as well as to fibrosis [5,6,34] and finally to a decline in renal function. A previous study showed that extracellular acidosis indeed leads to an inflammatory and fibrotic response in proximal tubule cells (NRK-52E) and renal fibroblasts (NRK-49F) as long as they are maintained separately in monoculture [16]. By contrast, under co-culture conditions, the acidosis-induced detrimental alterations were prevented in NRK-52E cells. As it is known that renal tissue can also activate protective programs under stress, we wanted to identify putative mechanisms underlying the observed beneficial impact of co-culture as an experimental model of relevant renal cell crosstalk. The acidotic stress applied (pH 6.0) was derived from elaborate in vivo measurements, showing that renal tissue pH in rodents reaches values well below 6.5 in stressful conditions [7,8,9].

We applied a undirected approach and determined the effect of extracellular acidosis on the transcriptome of proximal tubule-derived NRK-52E cells and renal fibroblasts (NRK-49F) in co-culture. The absolute number of affected genes was higher for NRK-52E cells, the overlap of affected genes in both cell types was 41% for NRK-52E and 60% for NRK-49F cells and the direction of differential expression was opposite in the cell types for a substantial number. These results already indicate that acidosis exerts a cell-specific change in the transcriptome and therefore in the cellular answer to this stress condition. Bioinformatic analysis (canonical pathways and upstream regulators) proposed activation of the Nrf2 pathway as the most likely cytoprotective strategy of NRK-52E cells. This prediction concurs with the suggested role of Nrf2 in kidney homeostasis and pathophysiology [17,18,19]. Regarding NRK-49F cells, canonical pathway analysis predicted activation of the Nrf2 pathway, however, this was not confirmed by the upstream regulator analysis, weakening the bioinformatics prediction for this cell type.

The Nrf2 pathway is a cytoprotective mechanism activated under different stress conditions, the classical example being oxidative stress. Nrf2 is a transcription factor, modulating gene expression via antioxidant response elements. The resulting alterations of the transcriptome affect cell metabolism (e.g., pentose phosphate pathway), redox balance (NADPH supply), cell survival, differentiation and proliferation [20]. Enhanced Nrf2 activity can result from enhanced expression, phosphorylation (best studied at Ser40, but also at other sites) and translocation to the chromatin fraction [21,35].

We show that acidosis enhances the expression of Nrf2 mRNA and protein (especially the chromatin fraction) expression in NRK-52E cells under co-culture conditions but not in monoculture. Furthermore, the abundance of phosphorylated Nrf2 in the chromatin fraction was also increased. Thus, we were able to confirm the bioinformatics predictions of an acidosis-induced activation of the Nrf2 pathway. In addition, our results are in accordance with reports on acidosis-induced activation of Nrf2 in other cell types [24,25].

According to our data, this activation results mainly from enhanced Nrf2 expression in the chromatin fraction. We cannot exclude the contribution of an enhanced phosphorylation rate, because the relative increase in pNrf2 was larger as compared to Nrf2. Nrf2 phosphorylation is mediated by different kinases such as PKC, CK2, ERK1/2 or AMPK [20,21]. PKC and CK2 seem to be the most important for phosphorylation at Ser40. However, our data do not allow conclusions regarding the kinases involved. We will address this question in further studies.

Our validation experiments for NRK-49F cells confirm the prediction of the upstream regulator analysis, showing that acidosis did not activate the Nrf2 pathway in this cell type. We observed even a slight inhibition of the pathway, although we cannot appraise its cell biological significance. Thus, the crosstalk between NRK-52E and NRK-49F cells is required for the cell-specific activation of the cytoprotective Nrf2 pathway in NRK-52E cells but not in NRK-49F cells. It will now be important to identify the mechanism underlying the crosstalk between the two cell types as well as the reason for cell specificity of Nrf2 pathway activation.

As already mentioned, Nrf2 is an important regulator of the cellular antioxidant response. The pentose phosphate pathway (PPP) is a major building block of this response [20]. An important enzyme herein is transketolase (TKT), which acts at the interfaces of the oxidative and nonoxidative arm of the PPP and of the glycolytic pathway [36]. Furthermore, the Nrf2 pathway tightly regulates TKT expression, leading to the hypothesis of acidosis-induced expression of TKT in NRK-52E cells in co-culture [37]. We were able to confirm this working hypothesis at the level of TKT protein expression. Under monoculture conditions, TKT protein was not enhanced during acidosis, although we observed a slight increase in TKT-coding mRNA. Either the increase in mRNA was not sufficient to translate into protein or additional mechanisms inhibited translation. In any case, co-culture was required for enhanced TKT expression and a discrepancy between changes in mRNA and protein is not uncommon [19].

## 5. Conclusions

In summary, our results show that metabolic acidosis leads to an activation of the Nrf2 pathway in renal epithelial cells but not in fibroblasts in co-culture. The crosstalk between the two cell types is necessary for this cytoprotective cell-specific effect in NRK-52E cells. Furthermore, activation of the Nrf2 pathway translates into enhanced expression of Nrf2 target genes, shown as a proof of principle for TKT. At present, we only know that the crosstalk is necessary, but we do not know by which means the crosstalk occurs. This important question has to be addressed in future studies.

The development and progression of chronic kidney diseases come with an acidic milieu leading to inflammation and fibrosis [3]. From our results, we suggest that epithelial–fibroblast crosstalk exerts protection against acidosis-induced cell injury in epithelial cells [16] at least in part via the Nrf2 pathway. Better knowledge of these protective processes could help to improve therapeutic strategies for the treatment of CKD.

## Figures and Tables

**Figure 1 antioxidants-12-00412-f001:**
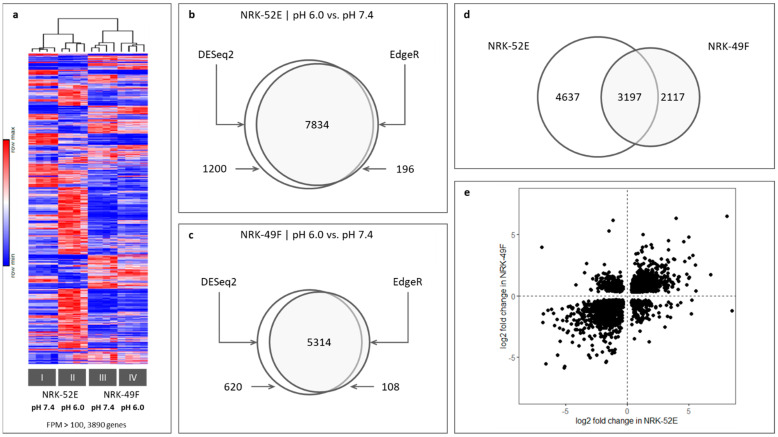
(**a**) Clustering of gene expression (https://software.broadinstitute.org/morpheus/; Euclidean distance, accessed on 22 November 2022) according to FPM values. Only genes with FPM > 100 are shown (48 h incubation). Clustering separates the samples clearly according to cell type and pH. (**b**,**c**) Overlap of differentially expressed genes by DESeq2 and edgeR analysis in NRK-52E and NRK-49F cells after 48 h exposure to acidic media. (**d**) Overlap of acidosis-induced differentially expressed genes in NRK-52E and NRK-49F cells (only genes identified by DESeq2 and edgeR were used). (**e**) Direction of overlapped differentially expressed genes in NRK-52E and NRK-49F.

**Figure 2 antioxidants-12-00412-f002:**
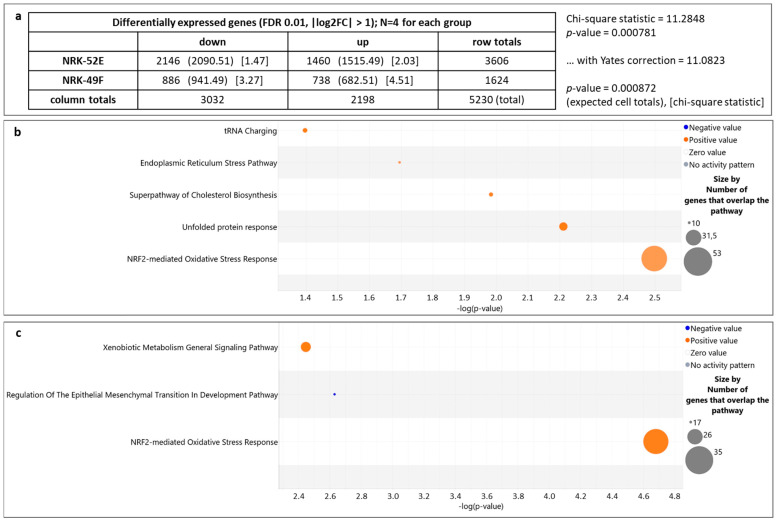
(**a**) Differentially expressed genes recognized by IPA with |log2FC| ≥ 1 in NRK-52E and NRK-49F cells. (**b**) Results of the canonical pathway analysis by IPA for NRK-52E. The Nrf2 pathway has the highest scores. (**c**) Results of the canonical pathway analysis by IPA for NRK-49F. The Nrf2 pathway has the highest scores.

**Figure 3 antioxidants-12-00412-f003:**
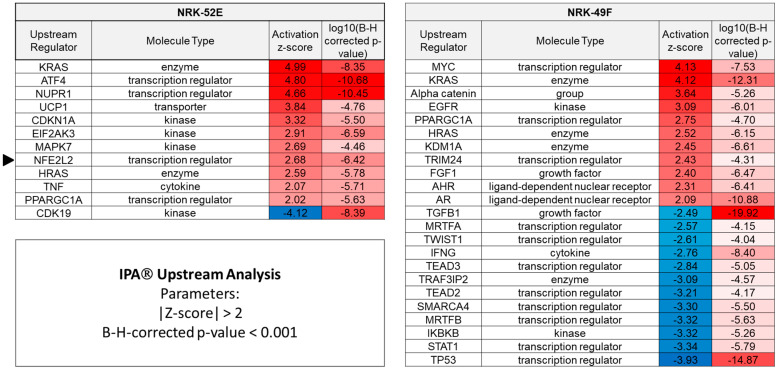
Results of the upstream regulator analysis by IPA for NRK-52E (**left**) and NRK-49F (**right**), applying higher stringency filters. The Nrf2 pathway (NFE2L2) is predicted for NRK-52E but not for NRK-49F cells.

**Figure 4 antioxidants-12-00412-f004:**
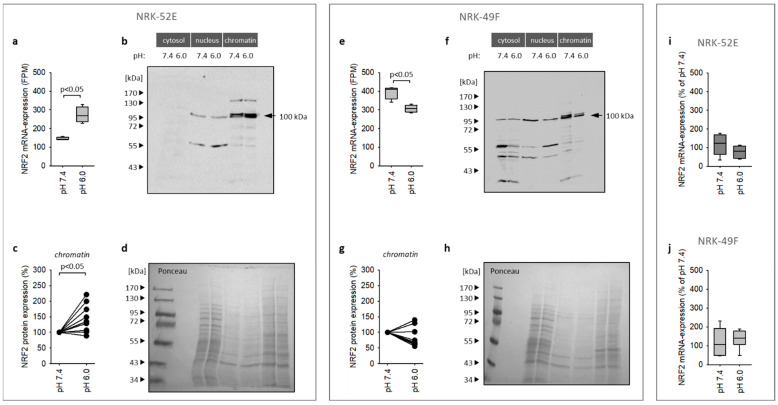
(**a**,**e**) Acidosis enhanced Nrf2 mRNA expression in NRK-52E cells in co-culture but not in NRK-49F cells. (*n* = 4). (**b**,**f**) Immunoblots for Nrf2 in three cell fractions of NRK-52E and NRK-49F cells. (**c**,**g**) Statistical analysis of Nrf2 expression in the chromatin fraction of NRK-52E (*n* = 9) and NRK-49F cells (*n* = 8). (**d**,**h**) Exemplary *Ponceau*-stained membranes (here for immunoblots (**b**) and (**f**)) show the protein loading of the gel and successful transfer to the membrane and were used for normalization. (**i**,**j**) Nrf2 mRNA expression was not enhanced by acidosis in monoculture (*n* = 6).

**Figure 5 antioxidants-12-00412-f005:**
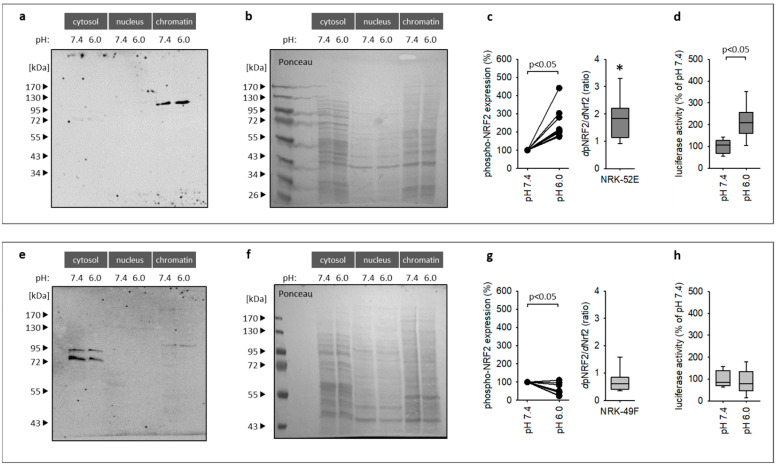
(**a**–**c**) Acidosis enhanced pNrf2^SER40^ expression to a larger extent than total Nrf2 (*d*pNrf2/*d*Nrf2) in the chromatin fraction of NRK-52E cells in co-culture. *n* = 9. * = *p* < 0.05 versus 1. (**e**–**g**) This was not the case for NRK-49F cells. (*n* = 8). (**d**,**h**) Acidosis enhanced the activity of the Nrf2 reporter gene luciferase in NRK-52E cells but not in NRK-49F cells. (*n* = 10). (**b**,**f**) Exemplary *Ponceau*-stained membranes (here for immunoblots (**a**,**e**)) show the protein loading of the gel and successful transfer to the membrane and were used for normalization.

**Figure 6 antioxidants-12-00412-f006:**
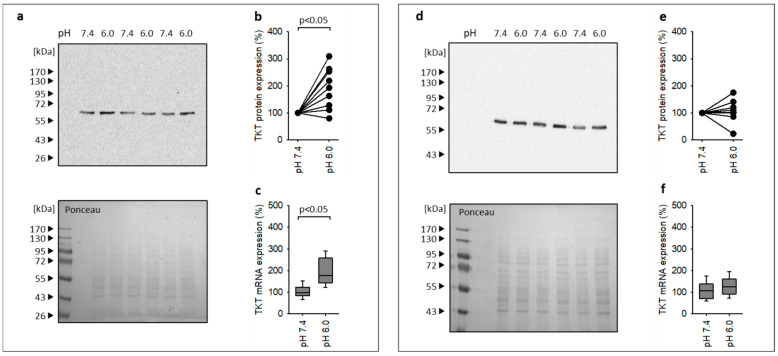
(**a**–**c**) Impact of extracellular acidosis on TKT mRNA and protein expression in NRK-52E cells under co-culture conditions. N for mRNA = 4. N for protein = 9. (**d**–**f**) Impact of extracellular acidosis on TKT mRNA and protein expression in NRK-52E cells under monoculture conditions. N for mRNA = 4. N for protein = 9. Exemplary *Ponceau*-stained membranes show the protein loading of the gel and successful transfer to the membrane and were used for normalization.

**Table 1 antioxidants-12-00412-t001:** Antibodies, order number, host and dilutions used.

Target	Company	Order Number	Host	Dilution
Antirabbit IgG HRP	Cell Signaling, Danvers, MA, USA	7074	Goat	1:1000
NRF2 (C-20)	Santa Cruz Biotechnology, Inc., TX, USA	sc-722	Rabbit	1:1000
Phospho-NRF2 (Ser40)	Thermo Fisher Scientific GmbH, Dreieich, Germany	PA5-67520	Rabbit	1:500
Transketolase	Cell Signaling, Danvers, MA, USA	E7O4M	Rabbit	1:2000

Abbreviations: HRP: horseradish peroxidase, NRF2: nuclear factor erythroid 2-related factor 2.

## Data Availability

The analyzed data supporting the conclusions of this article are included within this article and its additional files. Additionally, raw RNA sequencing data are publicly available in the Gene Expression Omnibus (GEO) database (https://www.ncbi.nlm.nih.gov/geo, accessed on 24 November 2022). GEO accession number: GSE218626.

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
