# Peer review of "Acidosis Activates the Nrf2 Pathway in Renal Proximal Tubule-Derived Cells through a Crosstalk with Renal Fibroblasts"

_antioxidants, 2023, doi:10.3390/antiox12020412_

Round 1
Reviewer 1 Report
The manuscript by Marie-Christin Schulz and collaborators describes a study aimed at understanding the molecular mechanisms underlying the process of protection from acidosis damage induced by the crosstalk between renal epithelial cells and renal fibroblasts. the study is based on a transcriptomic study and a Western Blot validation of the results performed on two cell lines grown in monoculture or co-culture.
The results obtained are interesting, even if the proposed study does not seem entirely in line with the purposes of this journal. however, there are some issues that should be resolved to make the manuscript suitable for publication.
1. A key point in the study of the molecular mechanisms of physiological and/or pathological processes is the appropriate choice of experimental conditions. in particular, the exposure time of the model under examination to a certain stimulus must be chosen rationally. The authors should clearly discuss how long the cells were exposed to acidosis conditions, why this exposure time was chosen, and whether an evaluation of the effect of exposure time on the transcriptome of the studied cells was made. 2. How did the authors verify there was no cross-contamination between the cells grown in coculture when they extracted the RNA or proteins? 3. The evaluation of the phosphorylation levels of a protein should always be expressed in relation to the total protein. the authors should therefore assess the variation of phsopho-Nrf-2 abundance versus absolute Nrf-2 abundance 4. The normalization of the densitometric analysis data of a WB by comparison of the ponceau staining of the filter is not reliable, even more so in this case in which the authors describe the dramatic effects of exposing cells to acidic conditions in terms of global variations of transcriptomics.
Reviewer 2 Report
The present study provides clear evidence of the need for the presence of at least two cell types in order to achieve a protective response to a potentially harmful change in the tissue (micro)environment. The authors confirm and refine their previous, rather recent study on the protective crosstalk of epithelial cells and fibroblasts in response to acidosis. This crosstalk integrates mechanisms that protect against inflammatory and fibrotic changes (up to serious and irreversible damage). In the present study the authors demonstrate the acidosis-induced Nrf2-pathway as one important module. They dissect the individual responses of the two cell types by analyzing their response to changes in the environmental pH either separately, i.e. for each cell type on its own (monoculture), or when the two cell types were kept in co-culture. Only the epithelial cells respond by a higher Nrf2 activity and the expression of the cytoprotective TKT, and only, when co-cultivated with fibroblasts. This finding should be of interest to a wide range of readers, because it may point to a universal principle, which could be transferred to many other organs/tissues with defined or derailed pH homeostasis. For example, bone homeostasis (osteoblasts, osteoclasts) or cancer tissue/stroma including the pre-metastatic niche, especially in adenomas/carcinomas evolving from epithelial tissues (cancer associated fibroblasts, cancer cells, tumor associated immune cells, ECM changes). The authors are free to include these thoughts in their Discussion or Conclusion. Another important finding is that bioinformatics can be very helpful when it comes to narrowing down possibilities and probabilities and making predictions, but by no means can bioinformatics replace experimental validation of a hypothesis, definitely not. This has been impressively shown and is then mentioned in the Discussion, e.g. page 10, ll. 20-22: “Regarding NRK-49F cells, canonical pathway analysis predicted activation of the Nrf2-pathway, however this was not confirmed by the upstream regulator analysis, weakening the bioinformatics prediction for this cell type.”
The paper is generally well written. Nevertheless, there are a few points and a number of little errors and typos the authors may want to check before having their manuscript published:
Minor:
- Page 2, paragraph 2.2., first two sentences are not clear enough. In the first sentence, the authors talk about “subsequent co-culture”. What is “subsequent” supposed to imply here? The second sentence lists two incubation periods, namely 24 h and 48 h. What makes the difference between the 24h and 48h? With (48h) and w/o (24h) serum? Were the cell lines kept separate during the first 24h and then co-cultured over the following 48h? The rationale behind the co-culture approach is well explained though.
- page 3, line 6, “… 4 µl glycogen and 180 µl and 99.9% ethanol”. I assume the authors meant to write “…and 180 µl of 99.9% ethanol.” However, a final volume is not given so that the amounts of 4 µl and 180 µl do not really help the reader to get the lay of the land in terms of the final EtOH concentration. On the same lines: can the glycogen concentration be given as µg/µl or mg/ml?
- Supplementary table. Composition of solutions: EDTA buffer: What was the counter(cat)ion for H2PO4-? Did the experimenter dissolve NaH2PO4 or KH2PO4?
- Page 5, line 14: Defining the FPM value as “Fragments per million mapped fragments” would be helpful to those who are not that familiar with this type of analysis.
- Page 11, line 5: “Under co-culture conditions, TKT protein was not enhanced during acidosis”. I am a bit confused here. Shouldn’t it read “In monoculture” or “Under monoculture conditions”, especially when recalling Figure 6?
Little errors and typos:
-Authors‘ affiliations, is the asterisk (first author) = correspondence (last author) placed correctly?
-Abstract, line 24: “Activation of Nrf2 comprises of enhanced….”, either remove “of” or replace “comprises” by “consists”.
-page 2, l. 9, “…was true as long AS the cells were…”
-page 2, l. 14, “…acts cytoprotectiveLY…”
-page 3, l. 3+4, “Turbo DNA-free kit” instead of “Turbo DNAse-free kit”?
-page 3, l. 8: “maximum possible value” (instead of maximal possible value).
-page 4, l. 3: “the pellet contain”ed” nuclear…” (instead of contains?).
-page 4, l.8: “ultrasonic”, is there a noun missing, or did the authors mean to write ultrasonic sound?
-page 5, l.6: punctuation mark missing between “SEM” and “Statistical”
-page 6, l.2: “IPA annota”t”ed 3603 genes“
-page 8, Figure legend 5, l.25: „…to a larger extend THAN total Nrf2….” (“than” instead of “as”).
-page 9, l. 2: “NRK-52EE”, one “E” to be removed.
-page 11, l. 9: Suggestion: “dissociation” could be replaced by “divergence” or “discrepancy”or “difference”.
-page 11, l.11: “5. Conclusions”, not bold?
-page 11, l. 14: “iS necessary”
-Reference #9, line 20: “…cou”n”tercurrent system…”
-----------------------------------------------------------------------
Round 2
Author Response
Response to the reviewers comments (round 2):
Comment 1 (These details have to be included into the manuscript.):
“The aim of this study is to reveal mechanisms, which are involved in the development of chronic kidney diseases. Therefore, we are committed to create an experimental setting, which approximates the pathological condition, as close as possible. This includes the longest possible incubation time, without critical decreases of cell viability. In a former study (doi: 10.3390/biomedicines10030681) we showed that incubation for 48 h is suitable. In addition, to ensure comparability of the results, we used the same experimental conditions in the present study.”
These details are now included in the Methods section.
Comment 2 (“can be excluded” sounds like a motivated hypothesis, but you cannot be sure, for example, that there are no cells grown on the wrong side of the filter ….”):
We explained the experimental setup in more detail in the Methods section. A cross contamination of cells is impossible by the setup (which is a frequently published standard setup for co-culture) used. A pore size of 0.4 µm, together with rinsing of the lower membrane side and inspection of the lower membrane side, is the established and accepted procedure in this case. https://pubmed.ncbi.nlm.nih.gov/35737233/
https://pubmed.ncbi.nlm.nih.gov/33281613/
https://pubmed.ncbi.nlm.nih.gov/33919334/
We describe these details now more extensively in the Methods section.
“Thus, in order to mimic the in vivo situation properly, NRK-52E cells were cultivated on the membrane of a transwell insert. NRK-49F cells were cultivated on the bottom of the wells of a 6-well-plate, in which the transwells were placed. There was no direct contact of the membrane with the bottom of the well. Due to the pore size of the membrane (0.4 µm) a contamination of the upper (transwell) compartment with fibroblasts or of the lower (6-well) compartment with epithelial cells is impossible. For processing at the end of the incubation period the transwells and the 6-well plate were separated, the lower side of the membrane as rinsed with ice-cold PBS and inspected to ensure that no material from the lower compartment was attached. Thereafter, the cells were processed separately. With these procedures we avoided any cross contamination of the two cell types.”
Comment 3 (“I apologize as my criticism was not clear enough. My intention was to ask you to insert an image that also shows a wb of Nrf-2 …”):
This is a new issue raised that was not part of the first review round and not raised by the other reviewer. Furthermore, Western blots for NRF-2 and pNRF-2 for both cells types were already included in the first version of the manuscript. Therefore, we do not understand the first sentence of this comment.
In summary, we cannot agree with the conclusion of the reviewer.
Detection of NRF-2 and phosphorylated NRF-2 at the protein level is known to be difficult, amongst others because it is a chromatin-attached transcription factor. As mentioned in the manuscript (page 8, lines 27ff.) we observed rather low levels of phosphorylated NRF-2 in NRK-49F cells, as compared to NRK-52E cells, making the detection even more difficult. This is explained in the manuscript.
The referred to WBs with ID 13, 18, 19, 23, 24, 12, 25, 26, are blots from NRK-49F cell preparations, with the mentioned low levels of pNRF-2. Of course, densitometric analysis is challenging but possible, because the background is low and homogenous. This is relevant for the signal to noise ratio and for setting the regions of interest around the bands. However, a higher variation has to be conceded. In sum, there was a slight negative effect of acidosis on pNRF-2 in NRK-49F cells and the pNFR-2/NRF-2-ratio was not affected significantly. Therefore, acidosis seems not to stimulate NRF-2 activity in NRK-49F cells. The NRF-2 reporter assay confirms this result. Thus, two independent experimental approaches support the same conclusion.
The referred to WBs with ID-15, 08, 03, 02, 01, 20, are blots from NRK-52E cell preparations, with a higher pNRF-2 band intensity in the chromatin samples from cells exposed to acidosis. The pNFR-2/NRF-2-ratio was also enhanced. Thus, there is evidence for acidosis-induced stimulation of NRF-2 activity in NRK-52E cells. The NRF-2 reporter assay confirms this result. Therefore, two independent experimental approaches support the same conclusion.
Comment 4 (“I don't agree with the authors, also in light of the low quality of the WBs. …):
In recent years, total cellular protein staining of the blotted membrane has become the preferred method for normalization of sample loading in Western blot analysis. This is documented by a large body of literature (see below). Especially for difficult samples, like cell fractionation or detection of chromatin-bound proteins, the use of reference proteins is problematic, because a variety of experimental conditions also affect these “housekeepers”. Therefore, the recommended procedure includes first the precise determination of the protein amount in the sample for loading (this ensures equal amount of protein per sample in the gel) and, second, the determination of total protein transferred to the membrane by a protein stain. We applied this procedure.
Publications regarding this topic:
https://www.sciencedirect.com/science/article/pii/S000326971930750X
https://www.sciencedirect.com/science/article/abs/pii/S0303720718300327
https://www.ncbi.nlm.nih.gov/pmc/articles/PMC3614345/
https://pubmed.ncbi.nlm.nih.gov/24023619/
https://doi.org/10.1002/pmic.201600189
https://www.sciencedirect.com/science/article/abs/pii/S0022175910000402
In a small internal series, using control whole cell lysate samples (i.e. without potential alteration of housekeeper expression by the experimental condition), we compared the Ponceau signal intensity with the common whole cell reference protein GAPDH to test the suitability under our technical conditions. The correlation was R2 = 0.9318, confirming the suitability of the Ponceau method, that will not be affected by experimental conditions.
